# Effect of CNT Content on Microstructure and Properties of CNTs/Refined-AZ61 Magnesium Matrix Composites

**DOI:** 10.3390/nano12142432

**Published:** 2022-07-15

**Authors:** Yunpeng Ding, Zhiai Shi, Zhiyuan Li, Sijia Jiao, Jinbiao Hu, Xulei Wang, Yizhuang Zhang, Hanying Wang, Xiaoqin Guo

**Affiliations:** School of Materials, Zhengzhou University of Aeronautics, Zhengzhou 450046, China; szhiai@126.com (Z.S.); lzy1260074530@outlook.com (Z.L.); jiaosijia@126.com (S.J.); jinbiao2021@126.com (J.H.); elan1220@163.com (X.W.); zyz11235@126.com (Y.Z.); hanyingwang@126.com (H.W.); glgcs@zua.edu.cn (X.G.)

**Keywords:** carbon nanotube, magnesium matrix nanocomposite, hot oscillatory pressing, mechanical property

## Abstract

Carbon nanotubes (CNTs) reinforced magnesium matrix composites have great application potential in the transportation industry, but the low absolute strength is the main obstacle to its application. In this paper, copper-coated CNTs and AZ61 powder were used as raw materials to prepare CNTs/refined-AZ61 composites with good interfacial bonding, uniformly dispersed CNTs and fine grains by the process of ball milling refinement of AZ61 powder, ball milling dispersion and hot-pressing sintering. When the volume fraction of CNTs is less than or equal to 1 vol.%, CNTs can be uniformly dispersed and the yield strength and compressive strength of composites increase with higher CNT content. When the volume fraction of CNTs is 1 vol.%, the compressive strength and yield strength of composites reach 439 MPa and 361 MPa, respectively, which are 14% and 9% higher than those of matrix composites with nearly the same value of fracture strain. When the volume fraction of CNTs is greater than 1 vol.%, with the increase in CNT content, CNT clustering becomes more and more serious, resulting in a decrease in the strength and fracture strain of composites.

## 1. Introduction

Against the background of energy conservation and CO_2_ emission reduction, magnesium alloys have great application potential in transportation, aerospace and 3C industries, due to the advantages of high specific strength, good dimensional stability and excellent machinability [1]. However, its development is restricted by low absolute strength, low modulus and poor creep resistance, so the reinforcements, such as SiC, Al_2_O_3_, TiC and carbon nanotube (CNT), are added in Mg alloys to fabricate magnesium matrix composites. Among these reinforcements, carbon nanotubes (CNT) are one of the most promising potential reinforcements [2] for magnesium matrix composites due to their advantages [3,4] such as high strength (~30 GPa), Young’s modulus (~1 TPa), ultra-high thermal conductivity (up to 6000 W/(m K)), electrical conductivity (~1.5 × 10^−6^ S/m) and high aspect ratio. With the same volume fraction of addition, the performance improvement of CNT is much better than that of commonly used SiC particle reinforcement [5]. The low absolute strength is still one of the main obstacles to the application of magnesium matrix composites [6]. In order to overcome this problem, CNT-reinforced magnesium matrix (CNTs/Mg) composites have become the focus of research.

Hou et al. [7] dispersed CNTs evenly and significantly refined Mg_17_Al_12_ phase through the processes of liquid dispersion, annealing, compacting, and extrusion, among others, fabricating MWCNTs/Mg-9Al composites with tensile strength of 355 MPa and fracture strain of 15%. Li et al. [8] introduced micro-Mg_2_Si and nano-Mg_2_Si through in situ reaction to CNTs/AZ91D composite, resulting in high mechanical properties. Abazari et al. [9] employed functionalized CNTs to enhance the interfacial bonding and prepared CNTs/Mg-3Zn composite with outstanding mechanical and biological properties.

The key factors affecting mechanical properties of CNTs/Mg composites include the alloy type and grain size of matrix, the content of CNT, the dispersion of CNT and the interface bonding. Therefore, as a common and inexpensive wrought magnesium alloy with medium level of strength and ductility, AZ61 was employed as the matrix in this paper. Furthermore, CNTs with metal coating were used to improve interfacial bonding. First, the matrix powder was refined by ball milling. Then, the dispersion of CNTs with AZ61 powder was improved by ball milling to prepare CNT-reinforced fine-grain AZ61 magnesium matrix (CNTs/refined-AZ61) composites with high strength and good toughness. The main focus of this study was the effect of carbon tube content on the microstructure and mechanical properties of the composites.

## 2. Materials and Methods

AZ61 magnesium alloy atomized powder (spherical, average particle size 50 micron, Nanou Co., Ltd., Shanghai, China) and copper-coated CNTs (length 5 μm, diameter 50 nm, copper content 50 weight %, prepared by electroless plating from Beijing Deke Daojin Technology Co., Ltd., China) were used as raw materials. The preparation process of composite materials mainly includes: ball milling refinement of AZ61 powder, ball milling dispersion of CNTs with AZ61 powder and vacuum hot-pressing sintering, as shown in Figure 1.

First, AZ61 powder was refined by the planetary milling process with zirconia balls. The parameters were as follows: ball milling speed 225 r/min, total milling time 30 h, ball material ratio 10:1. A total of 2% mass fraction of stearic acid was added to the powder as a control agent in order to avoid cold welding of powder during ball milling.

Then, a specific proportion (0.5, 1, 1.5 and 2 vol.%) of copper-coated CNTs, refined AZ61 powder were loaded into a ball mill tank with zirconia balls to prepare CNTs/refined-AZ61 composite powder. The mass ratio of zirconia ball to mixed powder was 10:1, the rotation speed was 225 r/min and the milling time is 4 h.

In the third step, the mixed powder was used to fabricate CNTs/refined-AZ61 composite by vacuum hot-pressing sintering (Centorr Vacuum Industries, Nashua, NH, USA). The heating rate was 10 °C/min. The temperature was raised to 450 °C and maintained at this level for 1 h to make the stearic acid in the mixed powder fully volatilized. Then, the temperature was raised to 500 °C and maintained at this level for an hour before furnace cooling.

Compression tests were performed on the composite samples with a size of 5 mm × 5 mm × 12.5 mm, using a mechanical testing machine (CMT5305, Suns, Zhangzhou, China) The compression speed was 0.0625 mm/min. Three compression tests were performed on each composite in the experiment, and the average result was taken as the final result of the compression test. The composite material was cut into small pieces, polished and corroded. Then, the metallographic microstructure was observed. The raw materials and composites were observed by scanning electron microscopy (SEM, JSM-7001F, JEOL, Tokyo, Japan). The preparation process of specimens for high-resolution transmission electron microscope (HRTEM) was as follows: The samples were cut into thin sheets with 0.2 mm thickness. Then, the sheets were sanded to 5–10 microns thickness using SiC sandpaper, followed by ion bombardment until the center was perforated. The interface structure of the composite was observed by transmission electron microscope (TEM, TECNAI G2 F20S-TWIN, FEI company, Hillsboro, OR, USA), and TEM images were analyzed and processed by Digital Micrograph software.

## 3. Results

### 3.1. Morphology and Microstructure

As shown in Figure 2a, the initial AZ61 powder is spherical with a diameter of about 80 μm. After a long period of ball milling, the AZ61 powder changed from spherical into irregular flakes with thickness of about 4 μm and length of about 20 μm, due to the friction and impact processes during ball milling [10], as shown in Figure 2b. These particles were formed by welding together smaller flake particles during milling. The refinement process of AZ61 powder increases the surface area, which is conducive to the dispersion of CNTs on the particle surface [11], and can improve the properties of the final material through fine-grain strengthening. Figure 2c,d show SEM diagrams of copper-coated CNTs. Most of the CNTs are covered with fine copper particles (<50 nm).

Figure 3 shows the SEM morphology of CNTs/refined-AZ61 composite powder dispersed by ball milling under different CNT contents. When the CNT content is smaller than 1%, CNTs are uniformly distributed across the surface of AZ61 powder (Figure 3a,b). This result shows that the Cu-coated CNTs could be evenly dispersed onto the surface of AZ61 particles by continuous violent collision, friction and stirring [10] during the ball milling process. However, when the content of CNT increases to 1.5 vol.%, clustered CNTs can be found on the surface of the mixed powder, as shown in Figure 3c. When the content of copper-coated CNTs is 2 vol.%, CNTs cluster with a size of about 4 μm in the mixed powder, as shown in Figure 3d. This is because the amount of CNTs is too much, and the ball milling dispersion can no longer disperse too many CNTs evenly, resulting in CNT clusters. Similar experimental results also appeared in other studies [3,12].

Figure 4 shows the microstructure of composites sintered with different CNT contents. It can be seen that most grains are thin strips with a length of about 20 μm and a width of about 4 μm, which is similar to the refined AZ61 particles (Figure 2b). Compared with the grain size of 80 μm in the original AZ61 powder, the grain size of the composite decreased significantly after ball milling. In addition, there are a small number of fine grains smaller than 3 μm, which are produced by the fine particles broken during the ball milling process (Figure 3). The grain morphology has little change with the difference of CNT content. However, with higher CNT content, the black spots in the microstructure increase, which corresponds to the CNT clusters in the mixed powder (Figure 3).

In order to further study the interfacial bonding of composites, HRTEM analysis was performed on the composites. Figure 5 shows HRTEM images of CNTs/refined-AZ61 composites. It can be seen that the interface between CNTs and Mg matrix is dense without pores, indicating that CNTs and AZ61 matrix achieve good interface bonding. In addition, the Digital Micrograph software was used to statistically measure the spacing of crystal planes. The spacing of stripes in Figure 5d is 0.34 nm, corresponding to CNTs, and the spacing of stripes around CNTs at 0.277 nm (Figure 5e) and 0.208 nm (Figure 5c) are corresponding to Mg and Cu, respectively. This also proves that copper particles (Figure 2d) on the surface of the original CNTs exist at the interface between CNTs and Mg matrix in the composites. This significantly improves the wettability of interface and makes it easier for the transfer of load from the matrix to the CNTs during deformation, thus strengthening the composites. The effect of Cu particles at the interface is similar to that of Ni around CNTs in the literature [13].

### 3.2. Mechanical Property

Figure 6 shows typical compressive stress–strain curves of composites with different CNT contents. Figure 7 shows the variation curves of compressive strength, yield strength and fracture strain of CNTs/refined-AZ61 composites with different CNT contents. It can be seen that the compressive strength and yield strength of composites have a consistent trend, which increases first and then decreases with the increase in CNT content. When CNT content is 1 vol.%, the mechanical properties of composites are the best, and the compressive strength and yield strength are 439 MPa and 361 MPa, respectively. Compared with pure AZ61, the compressive strength increases by 14% and the yield strength increases by 10%. However, the fracture strain hardly changes.

When the CNT content exceeds 1 vol.%, the strength of the composites decreases continuously with higher CNT content, even significantly lower than that of the matrix itself. This is because when the content of CNT is higher than 1 vol.%, CNTs cluster and fail to play the role of load transfer strengthening, but become the source of stress concentration and cracks [9,14], as well as pores, leading to a reduction in strength.

In addition, the toughness of composites gets increasingly worse with an increase in CNT content. During deformation, the reinforcement blocks dislocation movement and brings stress concentration, which leads to crack initiation and propagation. The higher the content of CNT, the easier the stress concentration, and the clusters of CNT at higher content add to this tendency [9], leading to fracture.

Table 1 shows a comparison between the performance of CNTs/refined-AZ61 composites prepared in this study and similar magnesium matrix composites reported in the literature [13,15,16,17]. In contrast, the composite prepared in this work has advantages in strength, especially in yield strength. In this study, the yield strength of the composites prepared only through sintering is much higher than the properties reported after further extrusion in other studies. Moreover, the grain size of matrix in this study is finer, compared with those reported in the literature.

### 3.3. Fracture Morphology

Figure 8 shows the fracture morphology of CNTs/refined-AZ61 composites. CNTs are obviously pulled out at the fracture surface in Figure 8a,b, indicating that CNTs played a bridging role in load transfer [18] and hindered the crack propagation during deformation.

With the increase in CNT content in composites, more and more CNTs could be observed at the fracture, as shown in Figure 8c. When CNT content is 2 vol.%, there are some large CNT clusters at the fracture surface, as shown in Figure 8d. Due to poor internal bonding force at the CNT cluster, it is easy for it to become the source of cracks due to stress concentration during deformation, resulting in cracking and the degradation of strength and fracture strain in composites [9]. This phenomenon corresponds to the compressive properties of composites.

## 4. Discussion

In the process of ball milling refinement, AZ61 powder refined from 80 μm to ~6 μm (Figure 2b), resulting in fine grains in the composite (Figure 4). This phenomenon results in a fine-grain strengthening effect, according to Hall–Petch relationship.

The copper-coated particles on the surface of CNTs (Figure 2c,d) have two functions. On the one hand, they reduce the density difference between reinforcement and matrix, which is conducive to dispersion during ball milling (Figure 4). On the other hand, they are beneficial to the interface bonding between CNTs and magnesium alloy [11].

When the volume fraction of CNTs is less than or equal to 1 vol.%, CNTs can be uniformly dispersed (Figure 3). Moreover, the copper coating of CNTs makes CNTs and AZ61 matrix achieve a better interface bonding (Figure 5), so that the strengthening effect of load transfer in CNTs can be fully used as the dominant strengthening mechanism [19] in CNTs/Mg composites. The uniform distribution of CNTs inhibits the early occurrence of cracks under static and dynamic loadings [9], and thus inhibits the reduction in fracture strain to some extent. In this study, the matrix structure is refined by using ball milling (Figure 4 and Table 1). As a result, compared with grain size of matrix reported in the literature (Table 1), this study’s is finer. Moreover, the Cu-coating of CNT makes the interface much stronger [4] as Ni-coating. Furthermore, the introduction of the ball milling process solves the problems of CNT dispersion (Figure 3) in CNTs/Mg composite. Thus, the properties (Figure 7) of the magnesium matrix composites are effectively optimized, resulting in better performance than the values found in the literature.

In short, CNTs/refined-AZ61 composites have excellent mechanical properties due to fine grains of matrix, good interfacial bonding and uniformly dispersed CNTs by the process of ball milling refinement of AZ61 powder, the introduction of metal coating around CNTs and the ball milling dispersion process.

## 5. Conclusions

In this work, the synthesis process and mechanical behavior of a refined-AZ61 matrix composite reinforced by uniformly distributed copper-coated CNTs are reported. The fabrication process included the following three steps: ball milling refinement of AZ61 powder, ball milling dispersion and hot-pressing sintering. With 1 vol.% volume fraction of CNTs, the compressive strength and yield strength of composites reached 439 MPa and 361 MPa, respectively, which are 14% and 9% higher than those of matrix. However, the fracture strain hardly decreases.

During the fabrication process, the size of matrix particles was reduced by the process of ball milling refinement, resulting the fine grains of matrix in composite. Furthermore, enhanced interfacial bonding via copper-plated CNT was used firstly in CNTs/Mg composites. Moreover, the dispersibility of CNT was improved by optimizing the ball milling process. Therefore, a better performance of composite was achieved in this study than in the literature values.

When the volume fraction of CNTs is less than or equal to 1 vol.%, the compressive yield strength and compressive strength of composites increase with higher CNT content, owing to the strengthening effect of more uniformly dispersed CNTs. When the volume fraction of CNTs is greater than 1 vol.%, with the increase in CNT content, CNT clustering becomes increasingly serious, resulting in a decrease in the strength and fracture strain of the composites.

## Figures and Tables

**Figure 1 nanomaterials-12-02432-f001:**
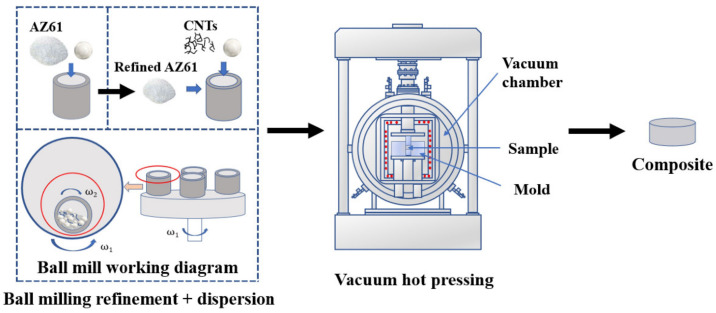
Preparation flow chart of composites.

**Figure 2 nanomaterials-12-02432-f002:**
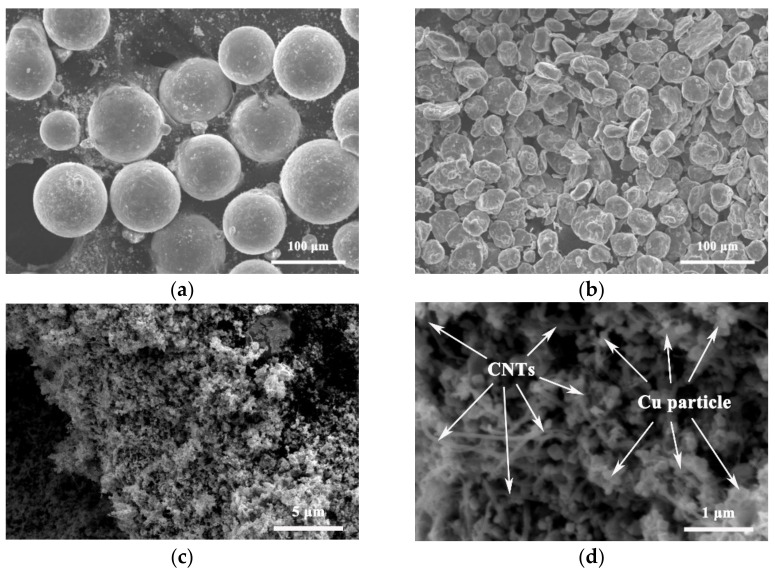
SEM images of AZ61 powder: (**a**) as-received and (**b**) after refinement of ball milling. SEM diagrams of copper-coated CNTs with (**c**) low magnification and (**d**) high magnification.

**Figure 3 nanomaterials-12-02432-f003:**
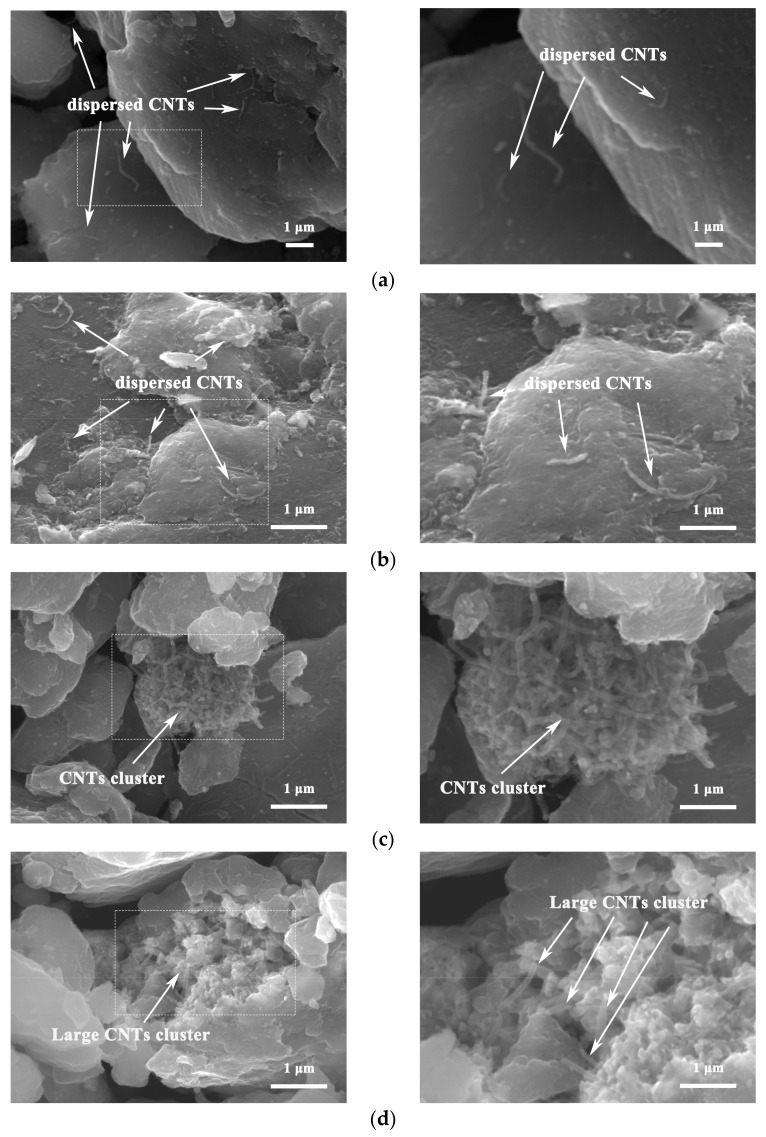
SEM morphology of CNTs/refined-AZ61 composite powder dispersed by ball milling under different contents of CNT. (**a**) 0.5%, (**b**) 1%, (**c**) 1.5%, (**d**) 2%.

**Figure 4 nanomaterials-12-02432-f004:**
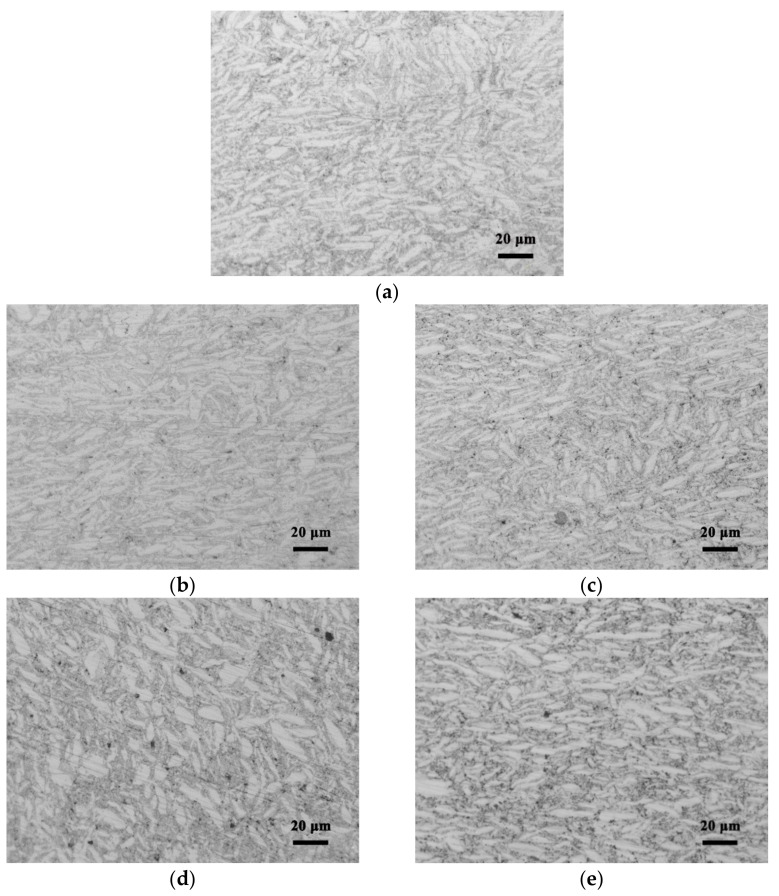
Microstructure of CNTs/refined-AZ61 composites with different CNT contents: (**a**) 0, (**b**) 0.5 vol.%, (**c**) 1 vol.%, (**d**) 1.5 vol.% and (**e**) 2 vol.%.

**Figure 5 nanomaterials-12-02432-f005:**
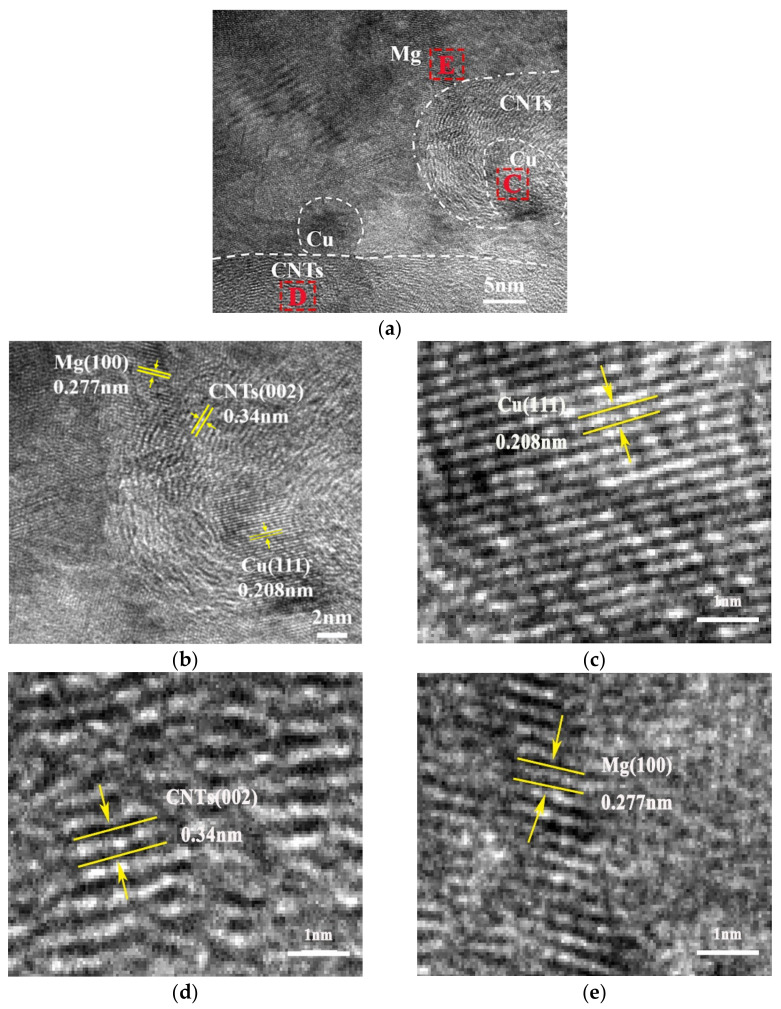
HRTEM diagrams of CNTs/refined-AZ61 composites with (**a**) low magnification and (**b**–**e**) high magnification from (**a**).

**Figure 6 nanomaterials-12-02432-f006:**
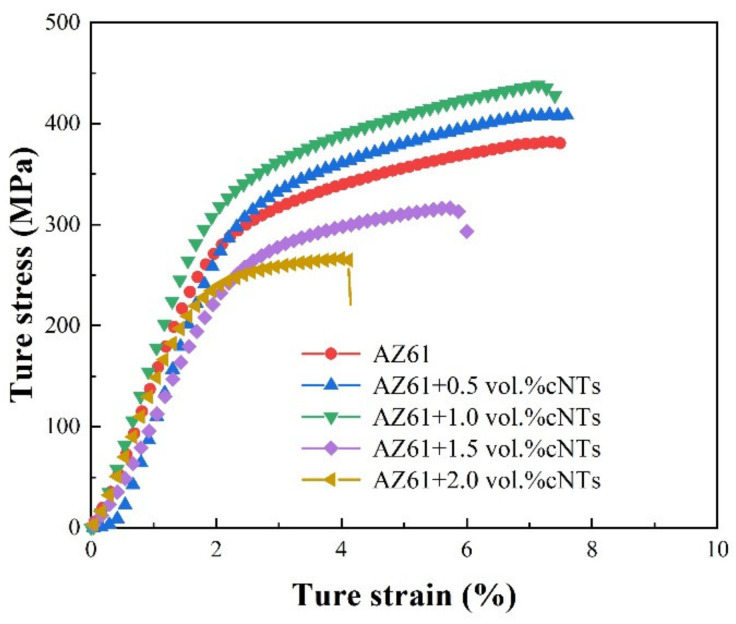
Typical compressive stress-strain curves of composites with different CNT contents.

**Figure 7 nanomaterials-12-02432-f007:**
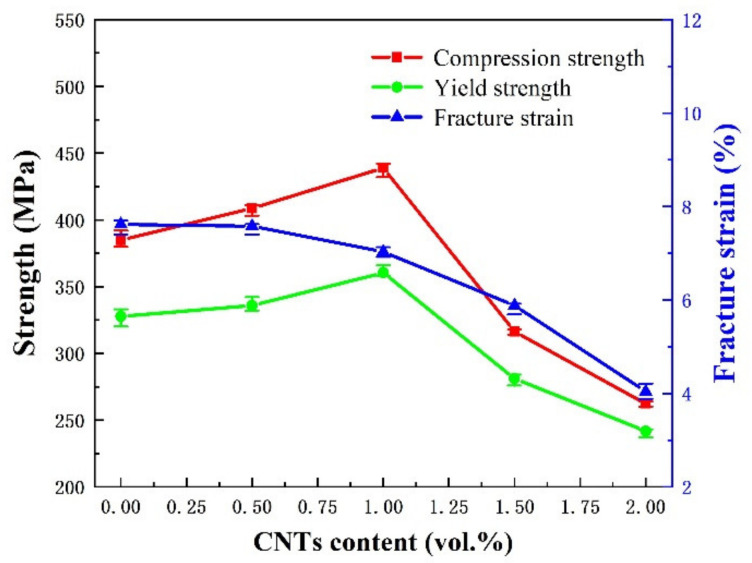
The variation curves of compressive strength, yield strength and fracture strain with different CNT contents in composites.

**Figure 8 nanomaterials-12-02432-f008:**
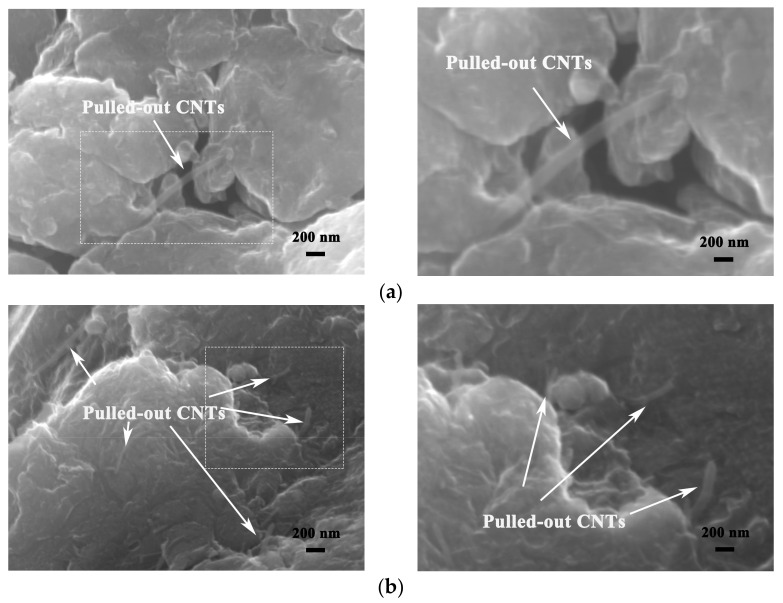
SEM images of fracture morphology in CNTs/refined-AZ61 composite with different CNT contents: (**a**) 0.5 vol.%, (**b**) 1 vol.%, (**c**) 1.5 vol.% and (**d**) 2 vol.%.

**Table 1 nanomaterials-12-02432-t001:** Comparison of compressive properties of magnesium matrix composites in the literature.

Matrix	Reinforcement	Preparation Process	Secondary Densification Process	Grain Size (um)	UCS/(MPa)	UTS/(MPa)	Ductility (%)	Ref.
AZ61	Cu-CNTs	Powder metallurgy	No	~6	361 (−2, +6)	439 (−7, +3)	7.0 (−0.11, +0.1)	This work
AZ31	Ni-CNTs + GNTs	Powder metallurgy	Hot extrusion	~50	198 ± 2.0	415 ± 1.8	11.8 ± 0.15	[11]
Mg	Al-CNTs	Microwave sintering	Hot extrusion	~10	144 ± 7	421 ± 11	11.3 ± 1.7	[13]
AZ81	CNT	Melt stirring	Hot extrusion	~8	129 ± 19	488 ± 13	16.0 ± 1.8	[14]
Mg	GNP + CNT	Powder metallurgy	Hot extrusion	~15	237 ± 4	425 ± 5	12.6 ± 0.2	[15]

## Data Availability

The results data are available to download from (https://pan.baidu.com/s/1_EF-9716-aaXoN_7UFkksg?pwd=x98e) with password of “x98e”. Accessed on 16 June 2022.

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
