# Peer review of "Effect of CNT Content on Microstructure and Properties of CNTs/Refined-AZ61 Magnesium Matrix Composites"

_nanomaterials, 2022, doi:10.3390/nano12142432_

Round 1
Reviewer 1 Report
This manuscript reports the microstructure and properties of magnesium AZ61 alloys refined by copper-coated carbon nanotubes (CNTs). AZ61-CNT composites were prepared by the ball milling method and hot-pressing sintering. They investigated the relationship between CNT contents and microstructure and mechanical properties. It was revealed that the best compressive strength and yield strength can be obtained with 1 vol.% of CNTs. This manuscript may include some important results, however, I do not recommend that it be published as it is due to the remarks given below.
[Remarks]
1. It seems that the background and motivation of this study in Introduction are scarce. It is necessary to explain the characteristics and current status of magnesium alloys, the current status and issues of magnesium alloys with reinforcing agents, and the reasons for choosing CNT.
2. While the authors state that carbon nanotubes are one of the ideal reinforcements, the problem is the low absolute strength of magnesium matrix composites, which seems contradictory. The research background should be described in more detail so as not to give such an impression.
3. Please explain why the authors chose AZ61.
4. Please explain how the copper-coated CNTs were prepared. In addition, please explain whether copper content 50% is weight%, volume%, or something else.
5. HRTEM photographs play an important role in this manuscript. Please add an explanation of how the specimens were prepared for HRTEM observation.
6. The authors prepared three compression test specimens, and the average values of their mechanical properties are shown in Figure 7 and Table 1. I am curious know how much variation the measured values have, so please exhibit the standard deviation or error bars.
7. It is difficult to recognize the CNTs indicated by the arrows in the SEM photographs of Figs. 3 and 7. I recommend that the authors use high magnification and resolution photos, or add element mapping by EDS.
8. The words "aggregate", "cluster" and "agglomeration" are confused. Unless there is a clear intention, they should be unified.
9. From the HRTEM in Fig. 5, it is difficult to understand that Cu exists at the interface between Mg and CNT and achieves excellent interfacial bonding. More detailed explanation is needed.
10. Table 1 shows the superiority of the compressive strength and yield strength obtained in this study compared with the literature values. I think it will be valuable to describe the mechanism of its superiority and difference from the literatures.
11. The conclusion is almost the same as the abstract. Please make a meaningful conclusion in connection with the microstructure found from this study.
Reviewer 2 Report
The article of Ding et al provides interesting results concerning microstructural and mechanical properties (mainly compressive strength and fracture strain) of CNT/AZ61 composites.
The main goal of this work is to investigate the effect of AZ61 alloys reinforcement by CNT particles.
The synthesis protocol of composite materials by ball milling is well described.
During their study the authors mainly focused on CNT and AZ61 particle size and on the CNT amount in the final composite material.
CNT, AZ61 and composites were characterized mainly by SEM images.
Mechanical properties associated with SEM measurements were described as a function of CNT amounts in composites. The authors have thus put forward the best composite formulation for optimized mechanical properties.
Composites characterization and mechanical properties are essentially based on SEM measurements. The goal of this work is reached namely the improvement of the mechanical properties.
In my opinion, all experimental results are well discussed.
I can’t find weaknesses.
I have just one minor recommendation: please provide Cu coated-CNT and AZ61 suppliers (in section 2).
Round 2
Reviewer 1 Report
This manuscript reports the microstructure and properties of magnesium AZ61 alloys refined by copper-coated carbon nanotubes (CNTs). This manuscript includes some important results and the authors revised the manuscript according to the reviewer's instructions and questions. I recommend that it be published in “nanomaterials”.